# Josephson junctions of topological nodal superconductors

Ranjani Seshadri,[1, *] Maxim Khodas,[2, †] and Dganit Meidan[1, ‡]

[1]*Department of Physics, Ben-Gurion University of the Negev, Beer-Sheva 84105, Israel*
[2]*Racah Institute of Physics, Hebrew University of Jerusalem, Jerusalem 91904, Israel*
(Dated: January 28, 2022)

Transition metal dichalcogenides (TMDs) offer a unique platform to study unconventional super-conductivity, owing to the presence of strong spin-orbit coupling and a remarkable stability to an in-plane magnetic field. A recent study found that when an in-plane field applied to a superconducting monolayer TMD is increased beyond the Pauli critical limit, a quantum phase transition occurs into a topological nodal superconducting phase which hosts Majorana flat bands. We study the current-phase relation of this nodal superconductor in a Josephson junction geometry. We find that the nodal superconductivity is associated with an energy-phase relation that depends on the momentum transverse to the current direction, with a $4\pi$ periodicity in between pairs of nodal points. We interpret this response as a result of a series of quantum phase transitions, driven by the transverse momentum, which separate a topological trivial phase and two distinct topologically non-trivial phases characterized by different winding invariants. This analysis sheds light on the stability of the Majorana flat bands to symmetry-breaking perturbations.

## I. INTRODUCTION

Recent advances in fabrications techniques have made it possible to engineer high-quality ultra-thin, multi-layer systems based on transition metal dichalcogenides (TMDs), with individual layers held together by weak Van der Waals forces [1, 2]. Some of these few-layered systems remain superconducting down to the monolayer limit [3–10]. Such TMD-based systems have been proposed as a platform for controlled studies of intrinsic or externally-induced unconventional superconductivity [11–26].

Unlike bulk systems, many TMD monolayers such as 1H-NbSe$_2$ lack an inversion center. This causes a spin-orbit splitting of electron bands, which polarizes the spin in the out-of-plane direction. The superconducting properties of such systems, termed Ising superconductors (SCs), [3, 5, 6, 11] are determined by the spin-orbit coupling (SOC), $\Delta_{SO}$ typically exceeding the superconducting gap by few orders of magnitude. In particular, Ising superconductivity is remarkably stable to the in-plane magnetic field, $\mathbf{B} \perp \hat{z}$. The in-plain critical field, $B_c$ greatly exceeds the Pauli limit [3, 5, 6, 8–10, 27] and at zero temperature is limited by the disorder [28–31].

While its presence explicitly breaks time-reversal (TR) symmetry $\Theta$, the stability of the superconducting state to the in-plane field can be understood as resulting from a modified TR symmetry $T$ which is a combination of the TR symmetry $\Theta$ and basal-plane mirror symmetry $M_z$ and is given by $T = M_z \Theta \tau_z$, where $\tau_{x,y,z}$ are the Pauli matrices in particle-hole Nambu space. This modified TR symmetry protects the superconducting state [15] and gives rise to field-induced triplet correlations [32].

Recently it was predicted that as the applied in-plane magnetic field exceeds the superconducting gap, a monolayer Ising SC transitions into a nodal topological SC [14]. The formation of nodal points is protected by the effective chiral symmetry for the particles moving perpendicular to the $\Gamma M$ line, which results from a combination of the modified TR symmetry $T$ and particle-hole symmetry. The nodal phase is expected to be accompanied by Majorana flat bands along the armchair edges [33, 34], experimental indications of which have been reported in [22, 35].

In this work we study the Josephson response of the nodal SC phase. We find that the nodal SC phase is associated with an energy-phase relation dependent on the momentum transverse to the current direction with a distinct $4\pi$-periodic Josephson current for the transverse momenta in-between the nodal points. We interpret this response as a consequence of a series of topological transitions resulting from the continuous change of the transverse momentum considered as the control parameter. The nodal momenta define the boundaries between a trivial phase and two topologically distinct non-trivial phases with different winding numbers. We further discuss the implications of these results on the stability of the Majorana flat bands in the presence of a symmetry-breaking perturbation.

The plan of this paper is as follows. We begin in Sec. II with a discussion of the symmetries that dictate the form of the low-energy Bogoliubov-deGennes (BdG) Hamiltonian of an Ising SC. In Sec. III we study the topological properties of the nodal SC phase and calculate the corresponding invariants. The current-phase relation in a Josephson junction made of two such nodal SCs is analyzed in Sec. IV and the results are presented in Sec. V, followed by a discussion of the underlying physical picture in Sec. VI. We accompany these qualitative arguments with a detailed description of edge states of an effective one-dimensional theory in appendix B, while the effect of the magnetic field on the transition is analyzed

---
\* ranjanis@post.bgu.ac.il
† maxim.khodas@mail.huji.ac.il
‡ dganit@bgu.ac.il

arXiv:2201.11514v1 [cond-mat.supr-con] 27 Jan 2022

in appendix A. The dependence of the Josephson current response on the junction barrier strength is briefly discussed in appendix C.

## II. BDG HAMILTONIAN FOR A NODAL SUPERCONDUCTOR

The band structure of TMD monolayers is constrained by the underlying crystalline symmetry with the point group $D_{3h}$. The symmetry operations include basal mirror reflection, $M_z$ which does not change the in-plane momentum and acts solely on the spin, $M_z = -i\sigma_z$. In addition, the rotation $C_3 = \left\{ e^{-i\frac{\pi}{3}\sigma_z} | \mathbf{k} \rightarrow \hat{R}_z(2\pi/3)\mathbf{k} \right\}$ around the $z-$axis acts both on spin and the momentum, with $\hat{R}_z(2\pi/3)$ being the spatial rotation by $2\pi/3$ around the $z-$axis. Finally, a vertical mirror passing via the high symmetry $\Gamma M$ line, taken here to lie along the $y-$axis, $M_x = \{-i\sigma_x | k_x, k_y \rightarrow -k_x, k_y\}$.

Inversion is not included in $D_{3h}$ group causing a finite Ising SOC. Thanks to the $M_z$ symmetry, the electron spins are polarized out-of-plane. The vertical mirror symmetry operation, $M_x$ makes the Ising SOC odd under the momentum reflection $k_x \rightarrow -k_x$. Correspondingly, the Ising SOC is even under $k_y \rightarrow -k_y$.

For definiteness, we consider the band structure of NbSe$_2$ monolayer with one band crossing the Fermi level. Different crossings give rise to the hole pockets centered at the $\Gamma$, $K$ and $K'$ points. The strong SOC near the $K(K')$ points protects the superconductivity in these pockets which is nearly unaffected by an in-plane magnetic field. Therefore, the analysis presented here relies solely on the presence of the $\Gamma$-point centered pocket(s) and applies equally to other systems such as gated MoS$_2$ [3, 5, 7], gated WS$_2$ [36], and metallic TaS$_2$ [9]. To understand the interplay between the SOC, in-plane field and superconductivity, it is sufficient to focus on the low-energy Hamiltonian as these energy scales are smaller than the typical Fermi energy.

To describe the spectrum of the $\Gamma$-pocket we expand the low energy single-band Hamiltonian,

$$H_0 = \xi(\mathbf{k})\sigma^0 + \lambda(\mathbf{k})\sigma^z. \tag{1}$$

up to the third order in k. Here $\xi(\mathbf{k}) = (k_x^2 + k_y^2)/2m - \mu$, where $\mu$ and $m$ are the chemical potential and the mass, respectively. The Ising SOC term is given by

$$\lambda(\mathbf{k}) = \lambda_I(k_x^3 - 3k_x k_y^2), \tag{2}$$

where $\lambda_I$ is the strength of the Ising SOC. As is clear from Eq.(2), this form of SOC vanishes along the $\Gamma M$ lines $k_x = 0, \pm\sqrt{3}k_y$. Note that in the following $ta^2 = m^{-1}$ is set to one, where $t$ is the tight binding hopping amplitude and $a$ is the lattice constant.

In order to study the interplay between an in-plane magnetic field $B$ characterized by the Zeeman energy $h = g\mu_B B/2$ applied for definiteness along the $x-$direction

and an $s$-wave spin-singlet superconducting pairing characterized by the gap, $\Delta$, we consider the BdG Hamiltonian,

$$\mathcal{H}(\mathbf{k}) = \xi(\mathbf{k})\tau^z + \lambda(\mathbf{k})\sigma^z + h\tau^z\sigma^x + \text{Re}(\Delta)\tau^y\sigma^y + \text{Im}(\Delta)\tau^x\sigma^y. \tag{3}$$

The magnetic field explicitly breaks TR symmetry $\Theta = \{i\sigma_y K | \mathbf{k} \rightarrow -\mathbf{k}\}$, where $K$ stands for complex conjugation. However, the combination of $\Theta$ and $M_z$ results in a modified TR symmetry, $T = \Theta M_z \tau_z = \{\sigma_x \tau_z K | \mathbf{k} \rightarrow -\mathbf{k}\}$, which squares to $T^2 = 1$. Moreover, thanks to the $M_x$ mirror symmetry, the Hamiltonian is even in $k_y$. We, therefore have the symmetry $T\mathcal{H}(\mathbf{k})T^{-1} = \mathcal{H}(-\mathbf{k}) = \mathcal{H}(-k_x, k_y)$ for the motion along $x-$direction when $k_y$ parameter is fixed, as is elaborated below.

The dispersion relation of the Bogoliubov quasi-particles inferred from Eq. (3) forms the four bands as shown in the Fig. 1 for selected values of the parameters ($m = 1$, $\mu = 0.25$, $\lambda_I = 0.15$, $h = 0.06$ and $\Delta = 0.02$). Figure 2 shows the lower positive-energy band (labelled 3 in Fig. 1) as a color plot.

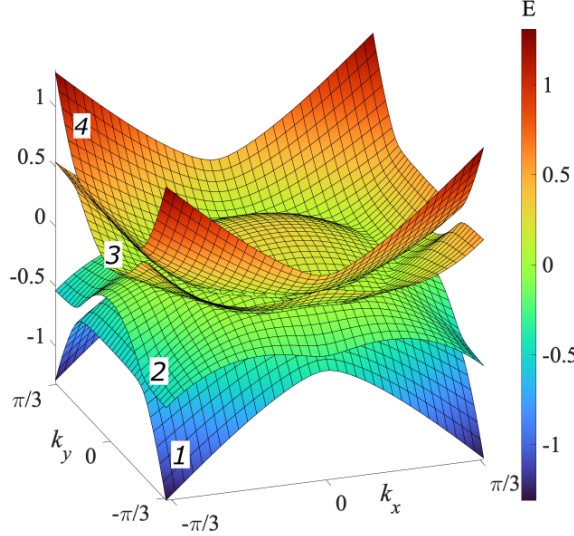

FIG. 1. The four band dispersion relation $E$ vs $\mathbf{k}$ of the Bogoliubov quasi-particles from the BdG Hamiltonian in Eq.(3). In this figure we have chosen $m = 1$, $\mu = 0.25$, $\lambda_I = 0.15$, $h = 0.06$ and $\Delta = 0.02$. The four bands are labelled $1 - 4$ in increasing order of energy. The surface plot of band 3 is shown in Fig. 2.

Due to the modified TR symmetry, superconductivity survives [15] when the magnetic field exceeds far beyond the Pauli limit. At $h = |\Delta|$, a quantum phase transition occurs, accompanied by the closing of the band gap at six discrete nodal points as detailed in the App. A. At yet stronger magnetic field exceeding the superconducting gap, $h > |\Delta|$ each nodal point splits in two, as shown by the red dots in Fig.2. The resulting twelve nodal points lie along high-symmetry $\Gamma M$ lines, marked in white.

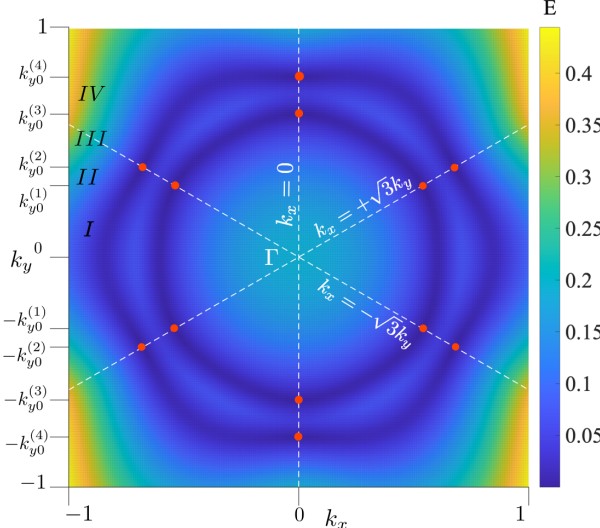

FIG. 2. Color plot of the lower positive energy band obtained (labelled 3 in Fig. 1). The spectral gap closes at twelve nodal points in the Brillouin Zone. These are shown as red dots and lie on the $\lambda(\mathbf{k}) = 0$ lines (white, dashed lines). The $y-$momenta $\pm k_{y0}^{(j)}$ mark the boundaries between the topological and non-topological phases which we cross as we change $k_y$ continuously.

We show below that this nodal SC phase, which arises from the model's non-trivial topological properties [14], is accompanied by a distinctive signature in the Josephson current response, which can be used as a probe to experimentally detect the topological phase.

## III. SYMMETRIES AND TOPOLOGICAL PROPERTIES

The origin of the nodal topological SC phase can be understood by considering the family of one-dimensional (1D) Hamiltonians obtained by setting the momentum $k_y$ as a parameter in the BdG Hamiltonian of Eq. (3). We denote this Hamiltonian as $\mathcal{H}_{k_y}^{1D}(k_x)$. While the magnetic field explicitly breaks TR symmetry, the family of 1D Hamiltonians has an emergent TR symmetry $\Theta$ such that $\Theta \mathcal{H}_{k_y}^{1D}(k_x)\Theta^{-1} = \mathcal{H}_{k_y}^{1D}(-k_x)$. In addition to the particle-hole symmetry $\Xi = \tau_x K$, the family of 1D Hamiltonians, $\mathcal{H}_{k_y}^{1D}(k_x)$ have a chiral symmetry $\mathcal{C} = \sigma_x \tau_y$ and fall under symmetry class BDI. As the parameter $k_y$ is varied, the gap closes and reopens at the spectral nodes. This closing and reopening of the gap is accompanied by a transition from a topologically trivial to a topologically non-trivial phase. To study the topological properties of this family of one-dimensional Hamiltonians, we rotate to the chiral basis $\tilde{\mathcal{H}}_{k_y}^{1D} = U\mathcal{H}_{k_y}^{1D}U^{-1}$ with $U = e^{-i\pi/4\tau_y}e^{-i\pi/4\sigma_x\tau_z}$.

The rotated Hamiltonian $\tilde{\mathcal{H}}_{k_y}^{1D}$ can be written as

$$\tilde{\mathcal{H}}_{k_y}^{1D}(k_x) = \begin{pmatrix} 0 & \mathcal{Q}_{k_y}(k_x) \\ \mathcal{Q}_{k_y}^\dagger(k_x) & 0 \end{pmatrix} \tag{4}$$

where:

$$\mathcal{Q}_{k_y}(k_x) = \begin{pmatrix} \xi_{k_y}(k_x) & h + i\lambda_{k_y}(k_x) - \Delta \\ h - i\lambda_{k_y}(k_x) + \Delta & \xi_{k_y}(k_x) \end{pmatrix} \tag{5}$$

with $\xi_{k_y}(k_x) = \frac{k_x^2}{2m} - \mu_{k_y}$, with $\mu_{k_y} = \mu - \frac{k_y^2}{2m}$ and $\lambda_{k_y}(k_x) = \lambda_I k_x(k_x^2 - 3k_y^2)$ and $h = \frac{1}{2}g\mu_B B$.

The topologically non-trivial phase of $\mathcal{H}_{k_y}^{1D}$ is associated with a winding of the phase of the determinant of the $\mathcal{Q}$ matrix: [37–41]:

$$\begin{aligned} W &= \frac{1}{2\pi}\int_{BZ_{1D}} \partial_{k_x} \mathrm{Im}\log \frac{\det(\mathcal{Q}_{k_y}(k_x))}{|\det(\mathcal{Q}_{k_y}(k_x))|}, \\ &= \frac{1}{2\pi}\int_{BZ_{1D}} \partial_{k_x}\phi_{k_y}^{(1)}(k_x) + \partial_{k_x}\phi_{k_y}^{(2)}(k_x), \\ &= W_1 + W_2, \end{aligned}$$

where we have used the fact that the winding of the determinant can be expressed as the sum of the winding of two complex eigenvalues of the $\mathcal{Q}$ matrix:

$$q_{k_y}^{(1,2)}(k_x) = |q_{k_y}^{(1,2)}(k_x)|e^{i\phi_{k_y}^{(1,2)}(k_x)}. \tag{6}$$

The winding of the phase of the determinant given by Eq. (6) for different values of $k_y$ is shown in Fig. 3(b). For $k_{y0}^{(1)} < |k_y| < k_{y0}^{(2)}$ (region II) the phase winds by $-4\pi$ and for $k_{y0}^{(3)} < |k_y| < k_{y0}^{(4)}$ (region IV) the phase of the determinant winds by $+2\pi$.

## IV. JOSEPHSON JUNCTION

The nodal topological SC phase is characterized by a flat band of Majorana modes that form on the armchair edges of the sample [14]. Here we show that the non-trivial topology also results in a distinctive current-phase relation when the system is patterned into a Josephson junction.

To study the current-phase relation we consider a junction between two nodal SCs as shown in Fig. 4. The SCs on either side of the junction are described by a bulk Hamiltonian of the form given in Eq.(3). The superconducting pairing has the same amplitude $|\Delta|$, but differs on either of the junction by a phase, i.e.

$$\Delta(x) = \begin{cases} |\Delta|, & \text{for } x < 0 \\ |\Delta|e^{i\phi}, & \text{for } x > 0. \end{cases} \tag{7a}$$

In addition, there is a $\delta-$function barrier at $x = 0$

$$U(x) = U_0\delta(x). \tag{7b}$$

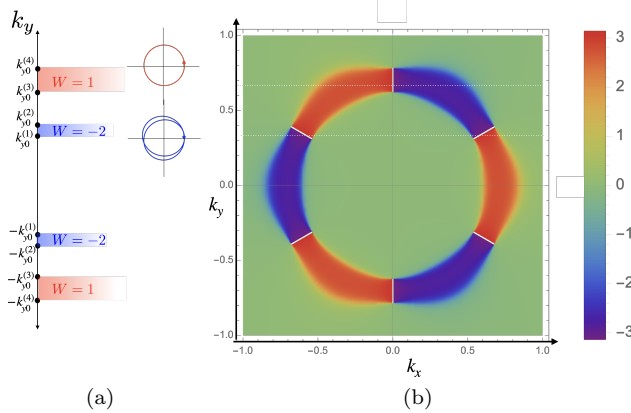

(a)                                         (b)

FIG. 3. (a) Phase diagram of the family of 1D chiral Hamiltonians as a function of the transverse momentum parameter $k_y$. The parameter range between the nodal points $k_{y0}^{(1)} < k_y < k_{y0}^{(2)}$ and $k_{y0}^{(3)} < k_y < k_{y0}^{(4)}$ corresponds to a topologically non-trivial phase with a topological invariant of $W = -2$ and $W = 1$, respectively. The winding $W$ of the phase of the determinant of the $\mathcal{Q}$ matrix, Eq. (6) accumulated as $k_x$ is swept over the Brillouin Zone is presented for the different values of $k_y$ in Fig (b).

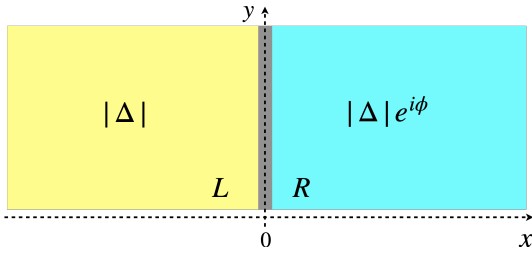

FIG. 4. Schematic showing the Josephson junction between two nodal SCs. The pairing $\Delta$ is in general complex and differs in phase between the two sides ($L$ and $R$) of the junction as given by Eq. (7a). The two sides are separated by a $\delta$−function barrier given in Eq. (7b).

Since the translational invariance along the $x$−direction is broken, we replace $k_x \to -i\partial_x$. However, since the translation symmetry along the $y$−direction is preserved, $k_y$ is still a good quantum number. Therefore, $k_y$ can be treated as a parameter and the Josephson-junction problem can be solved independently for each $k_y$. Written in this form, Eq. (3) becomes,

$$\mathcal{H}_{k_y}(x) = \left( -\frac{1}{2m}\partial_x^2 - \mu_{k_y} + U(x) \right)\tau^z$$

$$+ i\lambda(\partial_x^3 + 3k_y^2\partial_x)\sigma^z + H\tau^z\sigma^x$$

$$+ \text{Re}(\Delta(x))\tau^y\sigma^y + \text{Im}(\Delta(x))\tau^x\sigma^y \quad (7c)$$

The wave function $\Psi(x)$ for a given value of $k_y$ satisfies

the eigenvalue equation

$$\mathcal{H}_{k_y}(x)\Psi(x) = E_{k_y}\Psi(x), \quad (8)$$

with the Hamiltonian given by Eq. (7a) - (7c). Here $\Psi(x) = (u_\uparrow(x), u_\downarrow(x), v_\uparrow(x), v_\downarrow(x))^T$ is a four-component spinor, and we have suppressed the dependence of the wave function on the $k_y$ parameter for brevity. Let the wave function to the left and right of the barrier (labelled as "L" and "R" in Fig. 4) be denoted by $\Psi_L(x)$ and $\Psi_R(x)$ respectively. For the wave function to be continuous and differentiable at the junction, the following boundary conditions hold at $x = 0$

$$\Psi_L(0) = \Psi_R(0) \quad (9a)$$

$$\left.\frac{\partial\Psi_R(x)}{\partial x}\right|_0 - \left.\frac{\partial\Psi_L(x)}{\partial x}\right|_0 = 2\alpha\Psi(0) \quad (9b)$$

where $\Psi(0) = (\Psi_L(0) + \Psi_R(0))/2$ and $\alpha = mU_0$. Also, we define the transparency $D$ of the barrier as,

$$D = \frac{1}{\alpha^2 + 1} \quad (9c)$$

such that a transparent junction corresponds to $D = 1$, while an opaque barrier is given by $D \to 0$.

A particle with a given energy $E$ and transverse momentum $k_y$ propagates with linear momentum $k_x(E, k_y)$ which is determined by inverting the dispersion relation. In the four-band model we consider, for every pair of $E$ and $k_y$, there are four possible values of $k_x$ which in the presence of a superconducting gap are in general complex (i.e. have an oscillatory as well as a decaying component).

At energies below the induced superconducting gap $E < \Delta_{\text{gap}}$, the wave function decays exponentially away from the junction. Therefore, general solutions of Eq. (8) to the left/right of the barrier can be expressed as:

$$\Psi_L(x) = \sum_{j=1}^4 A_j \psi_{k_j} e^{ik_j x}, \quad (10a)$$

$$\Psi_R(x) = \sum_{j=1}^4 B_j \bar{\psi}_{k_j} e^{i\bar{k}_j x}, \quad (10b)$$

with $\text{Im}(k_j) < 0$ and $\text{Im}(\bar{k}_j) > 0$. Here each wave-function is a four-component spinor, i.e. $\psi_{k_j} = (u_{k_j,\uparrow}, u_{k_j,\downarrow}, v_{k_j,\uparrow}, v_{k_j,\downarrow})^T$.

Substituting the expression for the wave function in Eq. (10), the two boundary conditions Eq. (9a) and (9b) give us the following matrix equation for the coefficients $A_j$ and $B_j$,

$$MX = 0, \quad (11a)$$

where X is the column vector formed by $A_j$s and $B_j$s. The matrix $M$ is constructed using the four $k_j$ momenta and the corresponding four-spinors $\psi_{k_j}$ as follows,

$$M = \begin{pmatrix} \psi & -\bar{\psi} \\ -\psi(iK + \alpha\text{I}_{4\times 4}) & \bar{\psi}(i\bar{K} - \alpha\text{I}_{4\times 4}) \end{pmatrix} \quad (11b)$$

Here $\psi$ and $\bar{\psi}$ are the four matrices formed by the column vectors $\psi_{k_j}$ and $\bar{\psi}_{k_j}$ respectively. $K$ is a $4 \times 4$ diagonal matrix formed by the momenta $k_j$s, $K = \mathrm{diag}(k_1, k_2, k_3, k_4)$. Solutions to Eq. (11a) exist provided the determinant of $M$ vanishes. From this requirement we obtain the equation for the energies of the Andreev bound states as a function of the phase difference across the junction, i.e. $E_J(\phi)$. Using this we calculate the Josephson current as

$$I_J(\phi) = 2e \frac{\partial E_J}{\partial \phi} \qquad (12)$$

The periodicity of the energy $E_J$ and current $I_J$ tells us about the topological character of the system. While the non-topological regime shows a periodicity of $2\pi$, in the topological phase both quantities follow a $4\pi$ periodicity. This is explained in detail in the following section.

## V. RESULTS

We work in a parameter regime where the in-plane field exceeds the Pauli limit. In particular, we fix $h_x = 0.06$ and $|\Delta| = 0.02$. The spectrum, as noted earlier, is gapless and nodal points appear along $k_x = 0, \pm\sqrt{3}k_y$, i.e. the lines where the SOC vanishes. The dispersion relation for this set of parameters is shown in Fig. 2.

Figure 5(a) shows plots of $E_J$ vs $\phi$ along which the determinant of the matrix $M$ in Eq. (11b) vanishes for barrier transparency $D = 0.2$. This means that it is only along these curves that solutions for Eq. (11a) exist. The dark gray (dash-dot) and light gray (solid) curves belong to the non-topological regions I and III respectively. In both these regions we note that there is no zero-energy mode for any value of the phase $\phi$. Moreover, the periodicity of these two $E(\phi)$ curves is $2\pi$ as can be seen from the figure. On the other hand, the red (dotted) and blue (dashed) curves correspond to the two topological regions II and IV respectively. As is evident from the figure, these cross $E = 0$ at $\phi = \pi$ and $3\pi$ and have a periodicity of $4\pi$. Even though both these regions are topological, they differ in terms of the winding number as shown in Fig. 3. The blue (dashed) curve has two branches since the winding number in the corresponding region II $(k_{y0}^{(1)} < k_y < k_{y0}^{(2)})$ is $-2$; whereas the red (dotted) curve has only one branch as the winding number in region IV $(k_{y0}^{(3)} < k_y < k_{y0}^{(4)})$ is $+1$.

From this, we can also infer the behavior of the Josephson current $I_J$ using Eq. (12). This is shown in Fig. 5(b) as four panels for the four different regions. In the trivial regions I (dark gray, dash-dot) and III (light gray, solid) the current-phase relation has a period of $2\pi$. The current corresponding to the positive-energy branches is shown as a thick line whereas the negative-energy branch is shown as a thin line in both cases. On the other hand, both the topological regions II (blue, dash) and IV (red, dot) show a $4\pi$ periodicity in the current response.

The difference between these two phases is that while region II has two modes (corresponding to winding number $W = -2$), region IV has only one branch ($W = +1$). The thick (thin) curves in each panel correspond to the branches that have positive (negative) energy at $\phi = 0$.

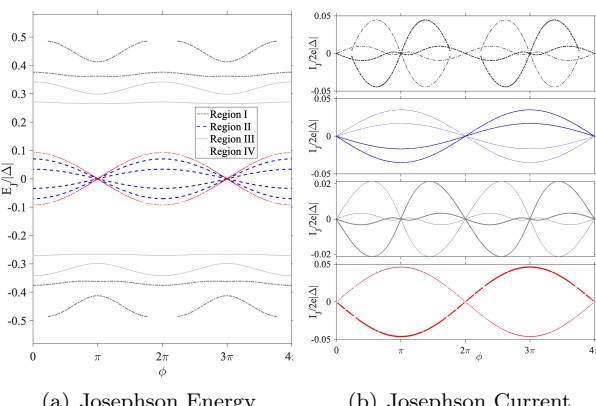

(a) Josephson Energy  (b) Josephson Current

FIG. 5. (a) Josephson Energy $E_J(\phi)$ and (b) Josephson Current $I_J(\phi)$ for the midgap states at representative points in the four different regions labelled $I - IV$ in Fig.2. The red (dotted) and blue (dashed) curves lie in the topological regions $II$ and $IV$ respectively. Clearly, these intersect the $E = 0$ line at $\phi = \pi$ and $3\pi$, and have a periodicity of $4\pi$. On the other hand, the dark gray (dash-dotted) and light-gray (solid) curves are far away from $E = 0$ for all values of $\phi$. These lie in the non-topological regions I and III respectively, where the zero-energy modes are not allowed. Here the periodicity is found to be $2\pi$. These observations are found to be true for various strengths/transparencies of the barrier separating the two sides of the Josephson junction.

As the transparency $D$ of the barrier is reduced, the mid-gap Andreev states in the trivial phase of the 1D Hamiltonian flatten and move to energies closer to the induced gap. On the other hand, for the set of transverse momenta that correspond to the topological regime, the presence of a pair of zero-energy Majorana modes at the barrier result in energy levels that stick to zero energy at a phase difference of $\phi = \pi$. This dependence on the barrier transparency is briefly explained in appendix C.

## VI. DISCUSSION AND PHYSICAL PICTURE

The behavior of the Josephson current can be understood qualitatively by studying the nature of the mid-gap states localized at each end of the junction in the weak-coupling limit. Here, the tunneling Hamiltonian between the two sides of the barrier can be treated in perturbation theory between otherwise decoupled half planes. Moreover, following the discussion in Sec. III each decoupled half plane can be treated as a family of semi infinite 1D wires governed by the Hamiltonian $\mathcal{H}_{k_y}^{1D}(k_x)$.

The underlying physical picture is depicted in Fig. 6 which shows the spin texture in the normal state ($\Delta =$

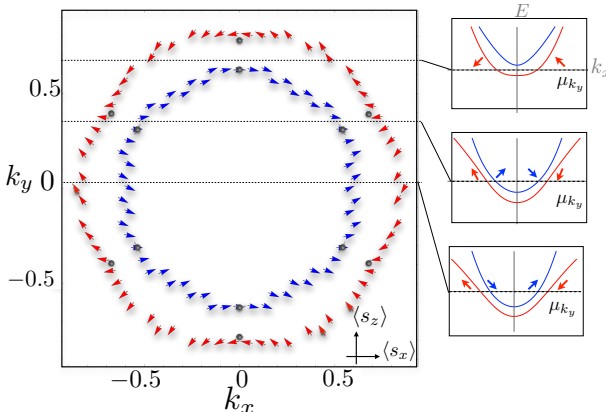

FIG. 6. The spin texture of the two Fermi surfaces as a function of $k_x$ and $k_y$. Line cuts at fixed $k_y$ show the dispersion of the two spin channels of the effective 1D model, corresponding to Region I, lower panel, Region II, middle panel and Region IV, upper panel.

0) of the two Fermi-surfaces split by in-plane field and SOC. Different cuts of fixed $k_y$ correspond to one/two populated bands with normal/flipped spin orientation.

In the nodal superconducting phase, the condition of $h \gg |\Delta|$ strongly suppresses inter-spin-channel pairing, and the induced superconductivity is predominantly intra-channel type. In region I and III corresponding to $|k_y| < k_{y0}^{(1)}$ and $k_{y0}^{(2)} < |k_y| < k_{y0}^{(3)}$ respectively, the chemical potential of the one-dimensional wire exceeds the Zeeman splitting $\mu_{k_y} = \mu - k_y^2/2m \gg h_x$ and the two spin channels acquire intra-band superconducting correlations with opposite winding numbers $W_{1,2} = \pm 1$, corresponding to a topological trivial phase. The two spin channels which exist as independent p-wave SCs in the bulk, are then coupled by the end of the wire. This coupling gaps out their midgap Majorana-like excitations resulting in a finite energy Andreev bound state on either side of the junction. Tunneling across the junction couples the Andreev states resulting in two modes with a $2\pi$-periodic energy-phase relation.

While the qualitative picture of two p-wave channels remains true for region II ($k_{y0}^{(1)} < |k_y| < k_{y0}^{(2)}$), in this parameter regime the spin orientation of the inner Fermi-surface is flipped with respect to region I, while the outer Fermi-surface remains unchanged. As a result the two p-wave spin channels have the same winding number $W_{1,2} = -1$. The two Majorana modes are protected by chiral symmetry and cannot be gapped by boundary. Consequently, in region II each half wire hosts two Majorana zero modes at its end. Treating the tunneling across the junction in perturbation theory would result in two mid-gap states with an energy-phase relation which is $4\pi$ periodic.

Finally, in region IV ($k_{y0}^{(3)} < |k_y| < k_{y0}^{(4)}$), the Zeeman splitting exceed the critical value $h > \sqrt{\Delta^2 + \mu^2}$ and the 1D wire is in the same topological class as that of the spin

orbit nanowire [42, 43]. Here only one of the two spin-channels is populated, leading to a single Majorana zero mode at either end of the junction. The tunneling across the junction would then lead to a single midgap state with $4\pi$ periodicity. The different scenarios are discussed in detail in App. B.

## VII. CONCLUSION

We study the current-phase relation of a nodal topological SC in a Josephson-junction geometry. Despite the presence of an in-plane field, the model retains an effective chiral symmetry which arises due to the particle-hole symmetry and a modified TR symmetry $T = M_z \Theta \tau_Z$. We find that the Josephson current-phase relation depends on the momentum transverse to the current direction and shows a distinctive $4\pi$ periodicity when the transverse momentum, treated as a control parameter, lies in-between pairs of nodal points (regions II and IV). The nodal momenta thus define the boundaries between a trivial phase and two topologically distinct non-trivial phases characterized by different winding numbers $W = 1$ and $W = -2$.

The $W = -2$ phase is protected by the chiral symmetry and is therefore unstable with respect to symmetry-breaking perturbations such as a Rashba SOC, which couple the two Majorana flat bands at the boundary. When such perturbations are present, the Josephson current-phase relation will exhibit a $2\pi$ periodicity in region $II$. Conversely, the $W = 1$ phase in region $IV$ is stable to weak perturbations that break the modified TR symmetry. We therefore expect that the $4\pi$ periodicity associated with this region will persist in the presence of weak symmetry-breaking perturbations.

## VIII. ACKNOWLEDGEMENTS

The authors would like to thank Beena Kalisky, Amit Keren and Binghai Yan for fruitful discussions. D.M. acknowledges support from the Israel Science Foundation (ISF) (grant No. 1884/18). M.K. and D.M. acknowledge support from the ISF, (grant No. 1251/19).

## Appendix A: In-plane field and phase transition

A remarkable property of nodal SCs is that the superconducting properties survive even when $h$, the applied in-plane magnetic field, is increased beyond the Pauli limit. At magnetic fields lower than the superconducting pairing, i.e. when $h < |\Delta|$ the spectrum is gapped at all momenta. As we increase the magnetic field, this energy gap reduces linearly as shown in Fig.7 (inset). This trend holds true for all $\Delta$s.

At $h = |\Delta|$ the system undergoes a phase transition which is accompanied by a closing of the spectral gap at

six points in the Brillouin zone (B.Z.). These six gapless points or 'nodes' lie at the vertices of a regular hexagon. These mark the points where the Fermi surface $|k_f| = \sqrt{2m\mu}$ intersects the $\lambda(\mathbf{k}) = 0$ lines.

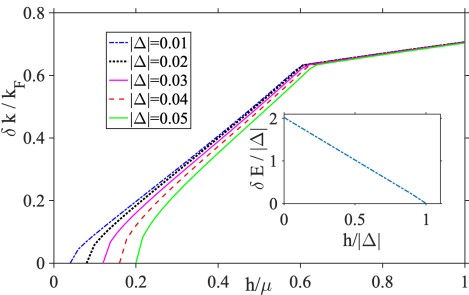

FIG. 7. Magnetic field-driven phase transition. The separation between each pair of nodal points increases as we increase the in-plane field beyond the superconducting pairing $\Delta$. This is shown for a range of $\Delta$s. The inset shows that the spectral gap $\delta E$ decreases linearly with increasing $h$. This is true for all $\Delta$s. The spectrum becomes gapless at $h = |\Delta|$ and continues to remain so for $h > |\Delta|$.

As $h$ is increased beyond the pairing $|\Delta|$ the spectrum continues to remain gapless. However, with increasing $h$, each of the six nodal points splits into two. This means that each B.Z. hosts a total of twelve nodal points which are shown in Fig.2. Each of the six pairs of nodal points continues to spread as $h$ is increased further. The separation of the nodes as a function of $h$ is shown in Fig.7 for a range of $\Delta$s.

## Appendix B: Edge states of the effective 1D Hamiltonian

We study the nature of the mid-gap states localized at each end of the junction in the weak-coupling limit, where, the tunneling Hamiltonian between the two sides of the barrier can be treated in perturbation theory between otherwise decoupled half wires. The underlying physical picture is depicted in Fig. 6 which shows the spin texture in the normal state ($\Delta = 0$) of the two Fermi surfaces split by in-plane field and SOC. Different cuts of fixed $k_y$ correspond to one/two populated spin-channels with normal/flipped spin direction. Different scenarios are discussed in detail below.

### 1. Region I: $|k_y| < k_{y0}^{(1)}$

This regime of parameters is characterized by two occupied spin-channels (see lower panel in Fig. 6). The condition, $\mu = k_F^2/2m \gg h, |\Delta|$ allows us to linearize the spectrum near the Fermi points. The general form of the

wave function is given by,

$$\psi_{\sigma=\uparrow,\downarrow} = R_\sigma(x)e^{ik_F x} + L_\sigma(x)e^{-ik_F x} \qquad (B1)$$

where $R_\sigma(x)/L_\sigma(x)$ are slowly varying functions of $x$ describing right/left moving electrons. In the basis of the slow varying fields $\psi_+ = (R_\uparrow(x), R_\downarrow(x), L_\uparrow(x)^\dagger, L_\downarrow(x)^\dagger)$ and $\psi_- = (L_\uparrow(x), L_\downarrow(x), R_\uparrow(x)^\dagger, R_\downarrow(x)^\dagger)$, the BdG hamiltonian is:

$$H_{k_y,\pm}^{1D}(x) = \mp iv_F \partial_x \tau_z + \pm\lambda k_F^3 \sigma_z + h\sigma_x \tau_z + \Delta \sigma_y \tau_y. \quad (B2)$$

We perform a rotation in the spin space to diagonalize the particle-conserving terms by applying the unitary transformation, $\tilde{H}_{k_y,\pm}^{1D}(x) = U^\dagger H_{k_y,\pm}^{1D}(x)U$, where $U = \exp\left[-i\sigma_y \tau_z(\pi/4 - \theta/2)\right]$ with the rotation angle $\theta$ defined via

$$v_F k_m \cos\theta = h$$
$$v_F k_m \sin\theta = \lambda k_F^3. \qquad (B3)$$

We find $\tilde{H}_{k_y,\pm}^{1D}(x) = H_\pm^0 + \delta H$ where

$$H_\pm^0 = \mp iv_F \partial_x \tau_z + v_F k_m \sigma_z \pm \Delta \sin\theta \sigma_y \tau_y$$
$$\delta H = \Delta \cos\theta \tau_x \qquad (B4)$$

and we have dropped the subscript $k_y$ for brevity. In the limit $h \gg \Delta$ we can treat $\delta H$ in perturbation theory. The unperturbed Hamiltonian $H_\pm^0$ admits two zero energy solutions at the end of the wire:

$$\psi_{\pm\downarrow}(x) = \frac{1}{\Omega}\begin{pmatrix} \beta e^{i\frac{\pi}{4}+i\frac{\phi(x)}{2}} \\ 0 \\ 0 \\ e^{-i\frac{\pi}{4}-i\frac{\phi(x)}{2}} \end{pmatrix} e^{\pm i(k_F - k_m)x}e^{\beta x/\xi}$$

$$\psi_{\pm\uparrow}(x) = \frac{1}{\Omega}\begin{pmatrix} 0 \\ \beta e^{-i\frac{\pi}{4}+i\frac{\phi(x)}{2}} \\ e^{i\frac{\pi}{4}-i\frac{\phi(x)}{2}} \\ 0 \end{pmatrix} e^{\pm i(k_F + k_m)x}e^{\beta x/\xi} \quad (B5)$$

where $\xi^{-1} = \Delta \sin\theta/v_F$ and $\beta = \pm$ for the left/right side of the junction, respectively, and

$$\phi(x) = \begin{cases} 0 & x < 0 \\ \phi & x > 0. \end{cases} \qquad (B6)$$

From these zero-energy solutions we can construct two zero-energy modes on each side of the barrier that satisfy the boundary conditions $\psi_M(x=0) = 0$ namely:

$$\phi_{\downarrow\beta}(x) = \frac{1}{\Omega}\begin{pmatrix} \beta e^{i\frac{\pi}{4}+i\frac{\phi(x)}{2}} \\ 0 \\ 0 \\ e^{-i\frac{\pi}{4}-i\frac{\phi(x)}{2}} \end{pmatrix} \sin[(k_F - k_m)x]e^{\beta x/\xi}$$

$$\phi_{\uparrow\beta}(x) = \frac{1}{\Omega}\begin{pmatrix} 0 \\ \beta e^{-i\frac{\pi}{4}+i\frac{\phi(x)}{2}} \\ e^{i\frac{\pi}{4}-i\frac{\phi(x)}{2}} \\ 0 \end{pmatrix} \sin[(k_F + k_m)x]e^{\beta x/\xi} (B7)$$

Calculating the matrix element due to the local inter-band pairing term, $\delta H$ between these states, we find that the each semi-infinite wire on either side of the barrier hosts a single Andreev end state with energy:

$$\beta\Delta_{gap} = \int dx \langle \phi_{M\uparrow\beta}(x)|\delta H|\phi_{\downarrow\beta}(x)\rangle$$
$$\approx 2\beta\frac{\Delta^3}{k_m^2}\cos\theta\sin^2\theta \qquad (B8)$$

Where we have used $k_F \gg k_m \gg \Delta$.

## 2. Region II, $k_{y0}^{(1)} < |k_y| < k_{y0}^{(2)}$

Once more the condition $\mu_y = \mu - k_y^2/2m = k_F^2/2m \gg h, \Delta$ is satisfied, corresponding to two filled spin bands. The situation in this regime is similar to that in region I with the spin orientation of the inner spin channel flipped while the outer Fermi surface remains unchanged, see Fig. 6 middle panel. The reason for the flip is that for the parameter range $k_y$ between the two nodal points $k_y = \frac{1}{2}(k_{y0}^1 + k_{y0}^2) \approx \sqrt{\frac{m\mu}{2}}$, the spin orbit term vanishes:

$$\lambda_I k_F(k_F^2 - 3k_y^2) = 0 \qquad (B9)$$

The magnetic field splits the two Fermi points by an amount $\sim \pm k_m$. As a result of this splitting, the two spin channels experience a finite SOC of opposite strength.

In the basis of the slow varying fields $\psi_+ = (R_\uparrow(x), R_\downarrow(x), L_\uparrow(x)^\dagger, L_\downarrow(x)^\dagger)$ and $\psi_- = (L_\uparrow(x), L_\downarrow(x), R_\uparrow(x)^\dagger, R_\downarrow(x)^\dagger)$ the BdG Hamiltonan is:

$$H_{k_y\pm}^{1D}(x) = \mp iv_F\partial_x\tau_z - i2m^2\lambda v_F\partial_x\sigma_z + h\sigma_x\tau_z + \Delta\sigma_y\tau_y \qquad (B10)$$

Next we will assume the eigenvectors have an oscillatory part and a slowly varying part:

$$\psi_{\pm s}(x) = e^{\pm sik_m x}e^{\beta x/\xi}\begin{pmatrix} R_\uparrow \\ R_\downarrow \\ L_\uparrow^\dagger \\ L_\downarrow^\dagger \end{pmatrix} \qquad (B11)$$

where $s = \pm$ correspond to the two spin eigenvalues.

The Hamiltonian in the 4-spinor basis becomes:

$$H_{k_y\pm,s}^{1D} = (sv_F k_m \mp i\beta v_F/\xi)\tau_z\sigma_0 \pm 2m^2\lambda v_F sk_m\tau_0\sigma_z$$
$$+ h\sigma_x\tau_z + \Delta\sigma_y\tau_y \qquad (B12)$$

performing a rotation in the spin basis $U = \exp\left[-i\sigma_y\tau_z(\pi/4 - s\theta/2)\right]$ this gives rise to $\tilde{H}_{k_y\pm,s}^{1D} = H_{\pm s}^0 + \delta H^1$:

$$H_{\pm s}^0 = (sv_F k_m \mp i\beta v_F/\xi)\tau_z + v_F k_m\sigma_z \pm s\Delta\sin\theta\sigma_y\tau_y$$
$$\delta H_s^1 = \Delta\cos\theta\tau_x \qquad (B13)$$

where $v_F k_m \cos\theta = h$ and $v_F k_m \sin\theta = m^2\lambda v_F k_m$. We therefore identify two zero energy solutions of the form

$$\phi_{\uparrow\beta} = \frac{1}{\Omega}\begin{pmatrix} 0 \\ \beta e^{-i\frac{\pi}{4}+i\frac{\phi(x)}{2}} \\ e^{i\frac{\pi}{4}-i\frac{\phi(x)}{2}} \\ 0 \end{pmatrix}\sin[(k_F+k_m)x]e^{\beta x/\xi}$$

$$\phi_{\downarrow\beta} = \frac{1}{\Omega}\begin{pmatrix} \beta e^{-i\frac{\pi}{4}+i\frac{\phi(x)}{2}} \\ 0 \\ 0 \\ e^{i\frac{\pi}{4}-i\frac{\phi(x)}{2}} \end{pmatrix}\sin[(k_F-k_m)x]e^{\beta x/\xi} \quad (B14)$$

crucially, the inter band pairing term does not couple the two zero mode:

$$\langle\phi_{\uparrow\beta}|\delta H|\phi_{\downarrow\beta}\rangle = 0. \qquad (B15)$$

Consequently, for this regime of parameters, each wire's end hosts two decoupled Majorana zero modes. Treating the tunneling Hamiltonian in perturbation theory [44] would give rise to two Andreev bound states:

$$E_{Js} = \Delta\sqrt{D_s}\sin\theta\cos\phi/2. \qquad (B16)$$

We note that the presence of the two Majorana modes is protected by the modified TR symmetry $T$, and is unstable to symmetry-breaking perturbation such as Rashba SOC, the presence of which would couple the two Majorana modes resulting in a single Andreev state on either side of the barrier. Consequently, when this $T$ symmetry is broken, this phase is continuously connected to region I and the nodal points along the $|k_x| = \sqrt{3}k_y$ line that separate the two will be gapped out.

## 3. Region IV, $k_{y0}^{(3)} < |k_y| < k_{y0}^{(4)}$

In this regime of parameters $h > \sqrt{\Delta^2 + \mu_{k_y}^2}$ and the effective 1D Hamiltonian is in the topological class of the case of the spin-orbit coupled nanowire [42, 43]. Here $\mu_{k_y} = \mu - k_y^2/2m$. In region IV only one of the two spin orbit bands is populated and each semi-infinite half wire hosts a single Majorana zero mode at its end. Treating the tunneling Hamiltonian in perturbation theory following ref. 44 would give rise to a single Andreev bound states:

$$E_J = \Delta\sqrt{D}\sin\theta\cos\phi/2. \qquad (B17)$$

Unlike region II, the presence of a single Majorana mode at either end of the junction remains stable to a weak symmetry-breaking perturbation.

## Appendix C: Dependence on barrier transparency $D$

The behaviour of Josephson energy $E_J(\phi)$ and current $I_J(\phi)$ depend not only on the $k_y$ cut we choose, but also

on the transparency of the barrier that separates the two sides of the junction shown in Fig. 4. We see that for a transparent barrier i.e. when $D \to 1$ the Josephson energies are close to zero at $\phi = \pi, 3\pi$ in all the regions. However, as we increase the strength of the barrier, i.e. as $D$ is reduced, the modes in the non-topological regions i.e. I and III are gapped out and move away from zero. However the modes in the topological regions II and IV continue to exist close to zero energy at $\phi = \pi, 3\pi$.

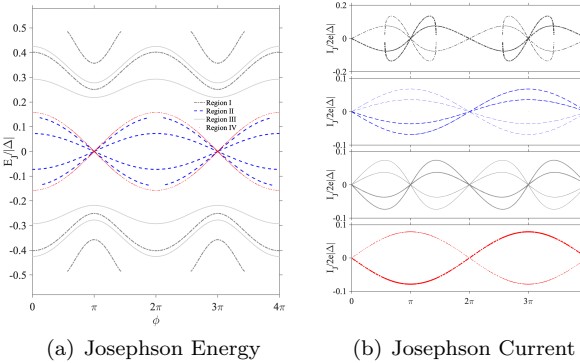

(a) Josephson Energy     (b) Josephson Current

FIG. 8. (a) Josephson energy $E_J(\phi)$ and (b) Josephson current $I_J(\phi)$ with barrier transparency $D = 0.6$. The colors correspond to the transverse momentum $k_y$ lying in the regions I − IV.

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
