# Peer review of "Josephson junctions of topological nodal superconductors"

_SciPost Physics_

## Round 1 · Referee Report · Anonymous · 2022-3-6

Strengths
The theory is solid.
The presentations are good.
Weaknesses
The theoretical method is limited. It may not apply to the real Josephson junctions. With a few analysis the results present in this work can be easily obtained, so the results are not quite new.
Report
The work by Seshadri, Khodas, and Meidan studied the Josephson junction formed in the monolayer superconducting NbSe2. Since it has been shown in previous works that an in-plane magnetic field can drive the monolayer NbSe2 to a topological nodal superconductor, the authors in this work focused on the current phase relation of the Josephson junction in the topological superconductor.
In this work the monolayer NbSe2 is treated as an effective superconducting wire characterized by a transverse ky momentum. The nodal points were found to separate the system into four regions, labeled by ky1, ky2, ky3, and ky4. The region I and III were found to be topologically trivial, so the phase current relation shows the standard 2\pi periodicity. The region II and IV were found to be topologically nontrivial, so the phase current relation shows the 4\pi period, indicating the Majorana zero modes host in the end of the wire. The region II and IV were further analyzed to show the difference brought by the different winding numbers in the two regions.
In my view point the theory of the work is solid and the presentations are good, so I could in principle recommend the publication of the paper. However, I think it is necessary to address the points raised in the below before the publication of the paper.
1) The way to treat the monolayer NbSe2 as a 1D effective wire characterized by ky seems convenient to solve the phase current relationship in the Josephson junction. However, in the real systems there is no real translation symmetry along y. In principle there are modes carrying ky from both the trivial and nontrivial regions. Can the authors comment how well can their results apply to the more realistic Josephson junction? If any simulation of a Josephson junction with finite width along y direction can be provided, then the quality of the work would raise a lot.
2) The Josephson current phase relation shown in Fig. 5 (b) is nice. The differences in the four regions are clearly shown. Is it possible to show a full evolution of the current phase relation from region I to the region IV? It would be helpful to the readers if the 2\pi to 4\pi transition can be shown when ky changes from one region to another region.
3) In the work the Josephson junction is assumed to be along x direction. What is the case when the Josephson junction is made along y direction or any other in-plane direction? Can the authors comment on how can their result be generalized to those cases?
There are a few minor points:
1) In the first sentence of the second paragraph in the introduction section, it is mentioned that the 1H-NbSe2 lacks an inversion centre. Is it 1H or 2H type structure?
2) Is it possible to choose a better view angle to show the nodal points in Fig. 1?
Author: Ranjani Seshadri on 2022-04-11 [id 2375]
(in reply to Report 1 on 2022-03-06)
We thank the referee for their comments and suggestions which helped us to improve the quality of our work. Our detailed response as well as the changes we intend to incorporate are detailed below.
-
In this work we have assumed that the electron momentum parallel to the junction is a conserved quantity. This allowed us to use $k_y$ as a parameter. As the referee has correctly pointed out, this treatment of a Josephson Junction is justified only when there is perfect translational invariance along the y-direction. However, in real systems the sample has a finite width. This would have two effects: 1) the confinement in the y-direction gives rise to quantization of $k_y$. If the width of the sample is $d_y$, ky is quantized as $\pi n/d_y$ where $n$ is an integer. 2) the open boundary condition will mix states with $\pm k_y $. The former would have the effect that only a discrete set of $k_y$ values are being sampled. As the topological regions II and IV occur in narrow strips in $k_y$, the effect would be noticeable if the junctions are wide enough such that the discretization is smaller than the splitting between the nodal points controlled by the in plane field. Regarding the latter effect, we note that the Hamiltonian, and consequently the winding numbers are symmetric under $k_y \rightarrow -k_y $. We therefore do not expect mixing between $\pm k_y $ to affect the results. We agree with the referee that the study of more realistic Josephson junction with finite width, orientation mismatch and disorder is an interesting topic which we intend to explore in a future work.
-
We thank the referee for this comment. We intend to include an animation that shows the full evolution with $k_y$ as supplemental material.
-
As we explain in the manuscript, the $4\pi$ periodicity can be understood as resulting from the appearance of Majorana zero mode at the two ends of the junction when the junction is oriented in the x-direction. We expect that, as is the situation with any nodal material (e.g. graphene), the edge states will not be present when the boundary orientation projects nodes of opposite charge onto each other, which in our case would be along the $y-$direction. Therefore we expect that the properties and stability of the topological regions II and VI will indeed depend on the orientation of the boundary with respect to the x-axis. In fact, to the extent that orientations can be experimentally controlled, we expect this to be a useful control knob that can be used to detect these effects. The effect of a general junction orientation with respect to the crystal on the Josephson response is an interesting topic which we intend to explore in a future work.
-
The system we consider here is a 1H-NbSe2 monolayer.
-
We have now modified Fig 1 such that the nodal points are visible more clearly.
Author: Ranjani Seshadri on 2022-04-11 [id 2373]
(in reply to Report 2 on 2022-03-15)We thank the referee for their comments and suggestions which helped us to improve the quality of our work. Our detailed response as well as the changes we intend to incorporate are detailed below.
The form of the barrier used in this work need not be non-superconducting. Only a weak link is required such that the phase change of the superconducting pairing across the junction is abrupt and not continuous. Josephson junctions in mono- or few-layer films can be realized by naturally occurring cracks in TMD flakes, see e.g. in https://arxiv.org/pdf/2106.11662.pdf, or by dynamically generated lines of reduced superconductivity, which arise due to the presence of screening currents generated by flux piercing see e.g. https://arxiv.org/pdf/2204.02888.pdf.
We address the effects of a finite junction width in our detailed response to referee A point 1. Regarding the possibility to control $k_y$: we note that one does not need to experimentally control $k_y$. Rather, for a wide and smooth junction, the Josephson current would be a sum of contributions from all transverse channels. In the nodal topological superconducting phase, some of these contributions would be $4\pi$ periodic which should be reflected in a fourier analysis of the Josephson signal. In addition, we expect that one way to control the relative contributions is by considering junctions at different orientations with respect to the x-axis. We intend to explore the detailed effect of junction orientation as well as other symmetry breaking perturbations which can be controlled experimentally in future work.
We have now added in the manuscript the following short summary of the method used to calculate the Josephson Energies. “To summarize, the Josephson energies are calculated using the following steps. Given an energy E and transverse momentum $k_y$, we invert the dispersion relation which results in four values of $k_x (E, k_y)$. Using these we construct the general wave function on either side of the barrier (Eqs. 10(a) and 10(b)) and match the boundary conditions at the junction, x = 0 (Eqs. 9(a) and 9(b)). This results in a matrix equation given in Eq. 11(a), which has solutions only when the determinant of M given by Eq. 11(b) vanishes. This condition gives us $E_J(\phi)$ for each $k_y$.” There is no simple analytical expression for $E_J(\phi)$
We have now edited the plots to be more readable and clearer than earlier.
This has been corrected
We have changed Fig 3 to explicitly indicate the different topological regions as a function of ky.

---

## Round 1 · Referee Report · Anonymous · 2022-3-15

Strengths
1. Concerns an important topic in condensed matter physics.
2. Proposes a way to probe nodal topological superconductivity in monolayer systems.
Weaknesses
1. Not fully clear, see comments below.
Report
In their work R. Seshardi and coauthors investigate signatures of nodal topological superconductivity in a Josephson junction defined in a monolayer transition metal dichalcogenide.
The paper introduces the concept of topological superconductivity by analyzing the symmetries of Ising superconductors in the presence of in-plane magnetic field and spin-orbit coupling that strongly polarizes the spins out of plane. The authors find regions in the momentum space, where the system is in topological regime. Finally they describe properties of a Josephson junction that dependently on the momentum in the direction perpendicular to the junction either has a trivial energy (current) vs phase relation with the period of 2$\pi$ or a topological one with the period of 4 $\pi$.
I find the message of the paper important to the field and I don’t have serious concerns regarding the performed study. I think the paper should be published after the authors clarify the following, minor issues.
Requested changes
1) Since the authors strongly emphasize the usage of Josephson junction as a test device for topological superconductivity please comment how such a monolayer junction, with non-superconducting barrier can be in practice realized.
2) What would be the effects of a finite width (with the width determining the junction span in the y-direction using the author’s nomenclature) of the junction. How in such a system the $k_y$ would be defined and controlled?
3) Please be more explicit on how $E_J$ is calculated. Can you provide a formula?
4) Fig. 5 is barely readable. I propose the authors to enlarge the plot, since it is the main result of the paper. Right now it is hard to see which curve is dashed, which is dotted.
5) I found a misspell in the second paragraph of the introduction. Instead of “in-plain” there should be “in-plane”. Also make explicit to what “its” refers in the first sentence of the third paragraph.
6) Can you explicitly denote the “regions” that you are referring thorough the paper in one of the maps that are plotted in $k_x, k_y$ space (e.g. Fig 3 or Fig. 6)?

---

## Editorial Decision

unknown